# Incidence, severity, and preventability of adverse events during the induction of patients with acute lymphoblastic leukemia in a tertiary care pediatric hospital in Mexico

Edmundo Vázquez-Cornejo[1]*, Olga Morales-Ríos[1], Gabriela Hernández-Pliego[2‡], Carlo Cicero-Oneto[2‡], Juan Garduño-Espinosa[3‡]

1 Department of Drug Assessment and Pharmacovigilance, Federico Gómez Children's Hospital of Mexico, Mexico City, Mexico, 2 Department of Hemato-oncology, Federico Gómez Children's Hospital of Mexico, Mexico City, Mexico, 3 Department of Research, Federico Gómez Children's Hospital of Mexico, Mexico City, Mexico

☉ These authors contributed equally to this work.
‡ GHP, CCO and JGE also contributed equally to this work.
* edmundoepiclin.qfb@outlook.com

## Abstract

Healthcare-associated adverse events represent a heavy burden of symptoms for pediatric oncology patients. Their description allows knowing the safety and quality of the care processes in countries with limited resources. This study aimed to describe the incidence, types, severity, and preventability of adverse events occurring in pediatric patients with acute lymphoblastic leukemia during the induction phase in a tertiary care pediatric hospital in Mexico. This study analyzed a cohort based on medical records of between 2015 and 2017. Initially, information on patients and adverse events was collected; subsequently, two pediatric oncologist reviewers independently classified adverse events, severity and preventability. Agreement between reviewers was evaluated. Adverse events incidence rates were estimated by type, severity, and preventability. One-hundred and eighty-one pediatric patients pediatric patients with acute lymphoblastic leukemia were studied. An overall adverse events rate of 51.8 per 1000 patient-days was estimated, involving 81.2% of patients during induction. Most adverse events were severe or higher (52.6%). Infectious processes were the most common severe or higher adverse event (30.5%). The presence of adverse events caused 80.2% of hospital readmissions. Of the adverse events, 10.5% were considered preventable and 53.6% could be ameliorable in severity. Improving the safety and quality of the care processes of children with acute lymphoblastic leukemia is possible, and this should contribute to the mitigation and prevention of adverse events associated morbidity and mortality during the remission induction phase.

**Data Availability Statement:** All relevant data are within the paper and its Supporting Information files.

**Funding:** The authors received no specific funding for this work.

**Competing interests:** The authors have declared that no competing interests exist.

## Introduction

Adverse events (AEs) are any injury to the patient related to medical management, in contrast to complications of an underlying disease [1]. Studying their frequency and impact in different areas and levels is considered necessary in order to adapt the processes of care, with the purpose to reduce harm and improve patient safety [2–4].

Medical management of cancer patients exposes them to the presence of unpleasant AEs related to the type of therapy they receive, which translates into a heavy burden of symptoms for them [5], particularly in pediatric oncology, where the frequency and severity of AEs can be underestimated [3]. It is also important to consider that chemotherapy protocols applied in low- and middle-income countries are usually adapted from successful protocols developed in high-income countries [6, 7], where health systems have more resources and conditions to facilitate access to structures and processes of care, and inequalities among its population do not pose a challenge of equal magnitude [8]. Therefore, it is necessary to investigate the safety of such adaptations in order to identify opportunities for reducing and preventing patient harm [6, 7].

An epidemiological description of AEs occurring in pediatric oncology care processes should include adverse drug events (ADEs), which are a common cause of hospitalization in children with neoplasms [9], as well as AEs related to medical procedures [2]. Moreover, an essential aspect is the determination of AEs preventability, as this informs about opportunities to improve patient safety in medical care processes [10, 11]. Preventability implies that the methods to avoid harm are known or that the harm is related to errors in medical care [12], so not all adverse events are preventable [4, 13]. However, a reductive perspective of preventability should be avoided, since not all errors in medical management lead to AEs [14].

In Mexico, the main neoplastic disease in childhood is acute lymphoblastic leukemia (ALL). Early mortality related to pediatric ALL treatment has not been reduced for decades [6, 15] and ranges from 5.4% to 15% during induction remission therapy [16, 17]. However, to the best of our knowledge, no description of AEs has been carried out in Mexico with a focus on processes of care and preventability in children with ALL, despite being important elements for patient safety. The scant evidence is on ADE related to chemotherapy medication errors (ME) [18], but the extent of AEs goes beyond drugs. Therefore, the purpose of this work is to comprehensively describe the incidence, type, severity, and preventability of AEs that occur in pediatric patients with ALL during the remission induction phase in a tertiary care hospital of Mexico City.

## Materials and methods

### Study design

Retrolective cohort study conducted in a single Pediatric National Health Institute in Mexico.

### Ethical approval

The HIM-2021-065 study protocol was submitted for review to the Research, Research Ethics, and Biosafety Committees of the Federico Gómez Children's Hospital of Mexico (HIMFG–Hospital Infantil de México Federico Gómez). The Research Ethics Committee evaluated the study design and the sources of information for this work, which involved a retrospective review of medical records generated during routine patient care and databases used for pharmacovigilance analysis, which is why this study did not involve the performance of medical interventions or intentional modification of physiological or psychological variables by the researchers on patients or their legal guardians. Therefore, the Institutional Research Ethics

Committee classified this project as research free of risk for the study subjects, with approval of the protocol being granted. In addition, the Research Ethics Committee exempted the researchers from obtaining informed consent, emphasizing the obligation of the researchers regarding the protection of the identity and privacy of the patients included in the study in order for sensitive information that might lead to patient identification not to be disclosed in this publication, in the supporting material or in the databases that are made available to the public for consultation.

In a previous peer-reviewed publication, we used information on chemotherapy from this cohort to study MEs [18]. We emphasize that the study focused on MEs and the present work on AEs have different purposes: the former focused on the quality of the chemotherapy medication process, and the current work focuses on the preventability of AEs related to all health-care-related interventions (drugs, procedures and hospital care). Since the primary variables in both studies are conceptually and operationally different, there is no overlap or duplication in the results. Finally, demographic data such as gender, age, risk classification, comorbidities and nutritional status shown in the previous publication have been included to contextualize the target population, without them being primary variables in the study.

## Setting and participants

HIMFG is a tertiary care pediatric hospital with teaching and research activities, with 229 registered beds, where one fourth of annual discharges are related to oncological conditions and the main morbidity taken care of is ALL [19]. To construct this cohort, all medical records of patients younger than 18 years, diagnosed with ALL between January 2015 and December 2017, were reviewed. All patients who received induction treatment at HIMFG until induction completion or patient death were included. Those patients with no information on therapeutic management or clinical evolution during remission induction were excluded.

## Medical management during remission induction

Every patient that comes to the HIMFG with suspected leukemia receives care to evaluate and stabilize their clinical status. The evaluation includes baseline lab tests, chest x-ray, testicular ultrasound, as well as diagnostic procedures such as bone marrow aspiration, lumbar puncture, cytogenetic analysis and immunophenotype of leukemia cells. When the diagnosis of ALL was confirmed, the HIM 2003 [20] protocol was started, which is adapted from St Jude Total 13 [21], with a therapeutic steroid window for seven days; when a lumbar puncture diagnosis is not possible, patients receive prednisone (60 mg/m$^2$/day), others use dexamethasone (6 mg/m$^2$/day). Subsequently, induction to remission is initiated with four weekly doses of vincristine (2 mg/m$^2$/dose), two weekly doses of daunorubicin (30 mg/m$^2$/dose), nine doses of L-asparaginase (10,000 UI/m$^2$/dose) that are interspersed in three weekly doses, 28 days with dexamethasone (6 mg/m$^2$/day), and four weekly doses of triple intrathecal chemotherapy. Patients with induction failure at day 21, receive cyclophosphamide (300 mg/m$^2$/dose) every 12 hours for three days. The evaluation of remission is performed with bone marrow aspirates at day 7, 14, 21 or 28; it is considered remission a bone marrow aspirate with <5% blasts at day 21. Remission in the nervous system is established with two consecutive negative lumbar punctures. It is considered that a patient is not in remission when blasts in bone marrow are >5% at the end of the first 4 weeks of induction.

## Study variables

**Type of adverse event.** The primary outcomes of this study are AEs occurred during remission induction therapy. An AE was defined as involuntary harm to the patient associated

with the medical care received, rather than complications caused by the primary disease [1, 22, 23]. Depending on their nature, a distinction was made between ADEs, AEs associated with medical procedures, and AEs associated with hospital care. ADEs were defined as noxious and involuntary responses to an appropriately or inappropriately used drug, i.e., appropriate dosage for prophylaxis, therapy or diagnosis (called adverse drug reactions [ADR]) or in the presence of MEs that cause harm to the patient [24]. A procedure-associated AE was defined as harm occurring during the technical execution of a planned procedure [25, 26] (e.g., lumbar punctures, central catheter insertion, bone marrow aspiration). AEs associated with hospital care were unintentional harm due to a failure to comply with management standards within the hospital [27] (e.g., hospital-acquired infections [HAIs]).

**Severity.** The degree of severity of each AE was defined according to the National Cancer Institute Common Terminology Criteria for Adverse Events (CTCAE), version 5 [28]. Five severity categories were included: grade 1 (mild), clinical or diagnostic observations, asymptomatic laboratory signs or mild symptoms that do not require intervention; grade 2 (moderate), requiring minimal, local or non-invasive interventions; grade 3 (severe), which require or prolong hospitalization or are disabling; grade 4 (life-threatening), when urgent medical intervention is required; grade 5 (death), when death is related to the AE. Each AE was accompanied by the system organ class assigned by CTCAE.

**Preventability.** The criteria to define preventability were adopted from the Gandhi scale [29]. An AE was considered preventable if it was the result of clinical care that was inconsistent with standard oncological practice, or a treatment-related complication that could have been anticipated according to current evidence [24, 29]. Those AEs whose severity or duration might have been considerably reduced if different actions had been taken were considered ameliorable, including those that are beyond current possibilities of hospital care [29, 30]. Those events that failed to meet any of the above criteria were considered non-preventable.

## Source of information

Patient information was extracted from medical records, which are archived on paper within the Department of Biostatistics. They are structured based on the Official Mexican Standard [31]. for medical records, and are prepared by resident and staff physicians, nurses, nutritionists, as well as by technical and administrative hospital staff. These records contain demographic, diagnostic, therapeutic information, medical evaluation notes, medical procedures, laboratory results, nursing care plan sheets and administrative documents.

## Data extraction

A structured review of medical records was carried out, where a trained pharmacist extracted demographic and diagnostic variables and data on clinical evolution, as well as complete information on therapeutic management during remission induction. To facilitate AEs identification, general triggering signals were used as a guide [32], but in order for induction chemotherapy-associated ADEs to be assessed as well, a synthesis of chemotherapy toxicity-related effects was prepared, which was used in the review of medical records.

In addition, the potential causes associated with suspected AEs were documented as exhaustively as possible, given the retrospective nature of the study, by capturing AE-related information from laboratory tests, specialty consultation notes, nursing notes, and evaluation notes, since the first hospital admission for diagnosis until the end of remission induction therapy or patient death. In case of death, the death certificate and, if available, the necropsy certificate was consulted.

During follow-up, the following variables were recorded: number of hospitalizations for AEs and their duration in days, as well as remission induction treatment duration. Similarly, information was collected on all prescribed medications (chemotherapeutic and non-chemo-therapeutic) at any hospitalization or outpatient services for each patient during follow-up, including the name of the medication, dose, administration route and frequency, in addition to administration initiation and conclusion dates.

## Classification of events

Once information extraction was completed, a detailed description of each patient's medical management was prepared, including causes and duration for hospital admissions, signs and symptoms indicative of suspected AEs, surgical procedures, and other outcomes in clinical course identified in the medical records, from diagnosis to follow-up conclusion in the cohort. In addition, chronological information on the pharmacological treatment administered with an evaluation of ME was included. Only for induction chemotherapy, deviations greater than 10% in the dose administered to the patient because of any failure in the medication process (prescription, transcription, administration) were considered dosing errors. This approach was adopted from other studies on medication errors in pediatric oncology [18, 33].

These descriptions were independently submitted to two expert pediatric oncologists who evaluated the information and confirmed or ruled out the presence of AEs, as well as associated causes. The level of agreement was evaluated as acceptable using the kappa test (k = 0.75). Each reviewer was then asked to assign the most appropriate CTCAE term for the AEs, as well as their degree of severity and preventability. Using the weighted kappa test ($k_w$), agreement between both reviewers was evaluated to assign the AEs degree of severity ($k_w$ = 0.68) and preventability ($k_w$ = 0.74). Disagreements between both reviewers were resolved by consensus in a virtual meeting with mediation of the researchers.

## Statistical analysis

AEs incidence rates were estimated by type, severity, and preventability per 1000 patient-days and per 1000 patient-days in hospitalization during induction. For this, total AEs and the number of in-hospital originated AEs were counted. Similarly, accumulated patient-days from the start of the diagnostic approach to the end of induction or patient death were counted, in addition to the number of patient-days spent in hospitalization. Additionally, the proportions of AEs by type, severity and preventability were estimated per 100 patients and for every 100 hospitalizations. For all estimators, 95% confidence intervals were calculated.

Univariate analysis of the information included categorical variables relative frequency and percentage (%), as well as quantitative variables central tendency and dispersion measures estimation according to their distribution. Incidence rates were tabulated by specific AEs, types, and causes. In addition, relative frequencies and percentages were tabulated grouped by severity and preventability. All analyses were carried out with the SPSS Statistics software, version 25.

## Results

In the study period, 207 patients were diagnosed with ALL. Among them, 181 (87.4%) with appropriate records of clinical evolution and interventions performed during the remission induction phase were included. Eleven patients (5.3%) were excluded due to medical records poor quality, as well as 15 patients (7.2%), who were referred to other centers to receive induction treatment. Mean age at the beginning of follow-up was 7.7 (± 4.5) years. At first admission, 18.8% of patients required intensive care for leukemia complications before the diagnosis

was established; subsequently, 13.3% required this type of care at some point during induction. The time required to establish a definitive diagnosis had a median of 3 (2–5) days. Mean cumulative hospital stay was 20 days, which accounted for 47.9% (± 28.1%) of total follow-up in this cohort (Table 1).

## Incidence of adverse events

In this cohort, 51.8 AEs per 1000 patient-days in the remission induction phase and 47.5 AEs per 1000 hospital patient-days were observed (Table 2). In total, 399 AEs were observed, involving 147 patients (81.2%). This meant an average of two AEs per patient during

**Table 1. Study population main characteristics.**

| Characteristics | Total (n = 181) |
|---|---|
| Gender–n (%) | |
| Females | 92 (50.8) |
| Males | 89 (49.2) |
| Age–n (%) | |
| Infants and neonates (0 to 23 months) | 15 (8.3) |
| Children (2 to 11 years) | 131 (72.4) |
| Adolescents (12 to 17 years) | 35 (19.3) |
| Immunophenotype–n (%) | |
| B | 167 (92.3) |
| T | 11 (6.1) |
| Mixed | 3 (1.6) |
| NCI risk classification–n (%) | |
| Standard | 64 (35.4) |
| High | 117 (64.6) |
| Leukocytes ($10^9$/L)–n (%) | |
| < 10 | 84 (46.4) |
| 10–49.99 | 57 (31.5) |
| 50–99.99 | 14 (7.7) |
| $\geq$ 100 | 26 (14.4) |
| Comorbidities–n (%) | |
| None | 162 (89.5) |
| $\geq$ 1 | 19 (10.5) |
| Nutritional status (z-score)–n (%) | |
| Adequate ($\geq$ -1.0 SD to $\leq$ +1.0 SD) | 88 (48.6) |
| Undernourishment (< -1.0 SD) | 38 (21.0) |
| Overweight (> +1.0 SD to $\leq$ +2 SD) | 34 (18.8) |
| Obesity (> +2.0 SD) | 21 (11.6) |
| Remission–n (%) | |
| Yes | 149 (82.3) |
| No | 23 (12.7) |
| Uncertain | 9 (5.0) |
| Follow-up time (days)—$\bar{x}$ (± SD) | 42.5 (±12.3) |
| Induction duration (days)—$\bar{x}$ (± SD) | 37.0 (±8.7) |
| Cumulative hospital stay (days)—$\bar{x}$ (± SD) | 20.0 (±12.6) |
| Mortality–n (%) | 20 (11.0) |

Abbreviations. NCI: National Cancer Institute, SD: standard deviation, $\bar{x}$: mean.

**Table 2. Incidence rates of adverse events during induction by type, severity, and preventability.**

| | Total | AEs per 1000 patient-days | | Total | AEs per 1000 patient-days in hospitalization | |
|---|---|---|---|---|---|---|
| | | Rate | 95% CI | | Rate | 95% CI |
| Overall | 399 | 51.8 | 46.7–56.9 | 172 | 47.5 | 40.4–54.6 |
| **Type** | | | | | | |
| Drugs | 367 | 47.7 | 42.8–52.5 | 140 | 38.7 | 32.3–45.1 |
| Hospital care | 19 | 2.5 | 1.4–3.6 | 19 | 5.2 | 2.9–7.6 |
| Procedures | 13 | 1.7 | 0.8–2.6 | 13 | 3.6 | 1.6–5.5 |
| **Severity** | | | | | | |
| Mild | 57 | 7.4 | 5.5–9.3 | 18 | 5.0 | 2.7–7.3 |
| Moderate | 132 | 17.1 | 14.2–20.1 | 58 | 16.0 | 11.9–20.1 |
| Severe | 128 | 16.6 | 13.7–19.5 | 46 | 12.7 | 9.0–16.4 |
| Life-threatening | 64 | 8.3 | 6.3–10.3 | 36 | 9.9 | 6.7–13.2 |
| Death-related | 18 | 2.3 | 1.3–3.4 | 14 | 3.9 | 1.8–5.9 |
| **Preventability** | | | | | | |
| Preventable | 42 | 5.5 | 3.8–7.1 | 36 | 9.9 | 6.7–13.2 |
| Ameliorable | 177 | 23.0 | 19.6–26.4 | 47 | 13.0 | 9.3–16.7 |
| Non-preventable | 157 | 20.4 | 17.2–23.6 | 86 | 24.0 | 18.7–28.8 |
| Non-evaluable | 23 | 3.0 | 1.8–4.2 | 3 | 0.8 | 0.0–1.8 |

Abbreviations. AE: adverse event, CI: confidence interval. Denominators: 7700 total patient-days and 3621 patient-days in hospitalization.

induction. One-hundred and seventy-two AEs (43.1%) were observed to occur during hospitalization (Rates per 100 patients or admissions are available in S1 Table).

In general, ADEs were the most frequent, accounting for 92.0% of total AEs (complete list in S2 Table). Specifically, 340 AEs (85.2%) were ADRs and 27 (6.8%) were MEs. The latter were mainly related to chemotherapy overdose and affected 22 patients (12.1%). The ME reached a rate of 7.5 [95% CI: 4.6–10.3] ME per 1000 hospital patient-days or 9.1 [95% CI: 5.8–12.3] MEs per 100 admissions. On the other hand, HAIs represented the main AE associated with hospital care. Together, they accounted for 4.3% of total AEs, and had incidence rates of 2.2 [95% CI: 1.2–3.3] HAIs per 1000 patient-days or 4.7 [95% CI: 2.5–7.0] HAIs per 1000 in-hospital patient-days. In addition, lumbar punctures were the main medical procedure causative of AEs, accounting for 2.8% of total (Table 3).

The most observed AEs were febrile neutropenia (18.8%), followed by allergic reactions (6.3%), hyperglycemia (6.3%), sepsis (6.0%), vomiting (6.0%), peripheral neuropathy (5.8%) and mucositis (5.0%) (Table 4). In general, blood and lymphatic disorders were the most widespread AEs, affecting 83 (45.8%) patients and reaching a rate of 12.3 AEs [95% CI: 9.9–14.8] per 1000 patient-days, followed by gastrointestinal disorders and infectious processes with rates of 9.6 [95% CI: 7.4–11.8] and 7.0 [95% CI: 5.1–8.9] per 1000 patient-days, respectively. However, whether all AEs involving infectious processes at any organic level are considered (including febrile neutropenia), these reached a rate of 17.4 [95% CI: 14.5–20.3] infections per 1000 patient-days. The incidence rate by system organ class is shown in S3 Table.

## Severity

There were 132 (33.1%) moderate, 128 (32.1%) severe, 64 (16.0%) life-threatening, 57 (14.3%) mild and 18 (4.5%) fatal AEs (Table 5). The incidence rate by severity grade is shown in

**Table 3. Adverse events incidence rates by cause during induction.**

| Cause | Total (n = 399) | AEs per 1000 patient-days | | Total (n = 172) | AEs per 1000 patient-days in hospitalization | |
|---|---|---|---|---|---|---|
| | | Rate | 95% CI | | Rate | 95% CI |
| **Drugs** | | | | | | |
| Adverse drug reaction[a] | 340 | 44.2 | 42.8–52.5 | 113 | 31.2 | 25.5–37.0 |
| Medication errors[b] | 27 | 3.5 | 2.2–4.8 | 27 | 7.5 | 4.6–10.3 |
| **Hospital care** | | | | | | |
| Hospital stay[c] | 10 | 1.3 | 0.5–2.1 | 10 | 2.8 | 1.0–4.5 |
| Catheter infection[c] | 5 | 0.6 | 0.1–1.2 | 5 | 1.4 | 0.2–2.6 |
| Surgical wound[c] | 2 | 0.3 | 0.0–0.6 | 2 | 0.6 | 0.0–1.3 |
| Others | 2 | 0.3 | 0.0–0.6 | 2 | 0.6 | 0.0–1.3 |
| **Procedures** | | | | | | |
| Lumbar puncture[d] | 11 | 1.4 | 0.6–2.3 | 11 | 3.0 | 1.2–4.8 |
| Catheter insertion | 2 | 0.3 | 0.0–0.6 | 2 | 0.6 | 0.0–1.3 |

Abbreviations. AE: adverse event. Denominators: 7700 total patient-days and 3621 patient-days in hospitalization.

[a]Includes the effects of toxicity associated with chemotherapy and other non-chemotherapeutic drugs.

[b]25 (6.3%) induction therapy dosing errors (> 10% overdose) are included, which affected 22 patients. In addition, two (0.5%) prescription errors that affected one patient with known allergy to acetaminophen are also considered.

[c]Hospital-acquired infections.

[d]Consequences of the procedure itself, rather than of the administered intrathecal chemotherapy.

Table 2. The incidence rates observed by type of AEs grade ≥3 in severity were: ADR (21.8/ 1000 patient-days [95% CI: 18.5–25.1]), MEs (2.2/1000 patient-days [95% CI: 1.2–3.3]), AEs associated with hospital care (1.9/1000 patient-days [95% CI: 1.0–2.9]), procedure-associated AEs (1.3/1000 patient-days [95% CI: 0.5–2.1]). Infections were the most common grade ≥ 3

**Table 4. Adverse events identified during induction therapy.**

| Adverse event | Adverse events (n = 399) | | Patients affected (n = 181) | | Incidence rates per 1000 patient-days | |
|---|---|---|---|---|---|---|
| | n | % | n | % | Rate | 95% CI |
| Febrile neutropenia | 75 | 18.8 | 67 | 37.0 | 9.7 | 7.5–11.9 |
| Allergic reaction | 25 | 6.3 | 20 | 11.0 | 3.2 | 2.0–4.5 |
| Hyperglycemia | 25 | 6.3 | 25 | 13.8 | 3.2 | 2.0–4.5 |
| Sepsis | 24 | 6.0 | 24 | 13.3 | 3.1 | 1.9–4.4 |
| Vomiting | 24 | 6.0 | 24 | 13.3 | 3.1 | 1.9–4.4 |
| Peripheral neuropathy | 23 | 5.8 | 23 | 12.7 | 3.0 | 1.8–4.2 |
| Mucositis | 20 | 5.0 | 18 | 9.9 | 2.6 | 1.5–3.7 |
| Platelet count decreased | 16 | 4.0 | 16 | 8.8 | 2.1 | 1.1–3.1 |
| Hospital-acquired infection | 10 | 2.5 | 9 | 5.0 | 1.3 | 0.5–2.1 |
| Anemia | 9 | 2.3 | 9 | 5.0 | 1.2 | 0.4–1.9 |
| Stroke | 9 | 2.3 | 9 | 5.0 | 1.2 | 0.4–1.9 |
| Seizure | 8 | 2.0 | 7 | 3.9 | 1.0 | 0.3–1.8 |
| Constipation | 7 | 1.8 | 7 | 3.9 | 0.9 | 0.2–1.6 |
| Electrolyte disturbance | 7 | 1.8 | 7 | 3.9 | 0.9 | 0.2–1.6 |
| Ileus | 7 | 1.8 | 7 | 3.9 | 0.9 | 0.2–1.6 |
| Neutrophil count decreased | 7 | 1.8 | 7 | 3.9 | 0.9 | 0.2–1.6 |

(*Continued*)

**Table 4.** (Continued)

| Adverse event | Adverse events (n = 399) | | Patients affected (n = 181) | | Incidence rates per 1000 patient-days | |
|---|---|---|---|---|---|---|
| | n | % | n | % | Rate | 95% CI |
| Abdominal infection | 6 | 1.5 | 6 | 3.3 | 0.8 | 0.2–1.4 |
| Hypertension | 6 | 1.5 | 6 | 3.3 | 0.8 | 0.2–1.4 |
| Abdominal pain | 5 | 1.3 | 5 | 2.8 | 0.6 | 0.1–1.2 |
| Catheter related infection | 5 | 1.3 | 5 | 2.8 | 0.6 | 0.1–1.2 |
| Epistaxis | 5 | 1.3 | 5 | 2.8 | 0.6 | 0.1–1.2 |
| Gastritis | 5 | 1.3 | 5 | 2.8 | 0.6 | 0.1–1.2 |
| Multi-organ failure | 5 | 1.3 | 5 | 2.8 | 0.6 | 0.1–1.2 |
| Pancreatitis | 5 | 1.3 | 5 | 2.8 | 0.6 | 0.1–1.2 |
| Skin infection | 5 | 1.3 | 4 | 2.2 | 0.6 | 0.1–1.2 |
| Cushingoid | 4 | 1.0 | 4 | 2.2 | 0.5 | 0.0–1.0 |
| Cerebrospinal fluid leakage | 3 | 0.8 | 3 | 1.7 | 0.4 | 0.0–0.8 |
| Disseminated intravascular coagulation | 3 | 0.8 | 3 | 1.7 | 0.4 | 0.0–0.8 |
| Hepatic failure | 3 | 0.8 | 3 | 1.7 | 0.4 | 0.0–0.8 |
| Superficial thrombophlebitis | 3 | 0.8 | 3 | 1.7 | 0.4 | 0.0–0.8 |
| Dysesthesia | 2 | 0.5 | 2 | 1.1 | 0.3 | 0.0–0.6 |
| Gastric hemorrhage | 2 | 0.5 | 2 | 1.1 | 0.3 | 0.0–0.6 |
| Hematuria | 2 | 0.5 | 2 | 1.1 | 0.3 | 0.0–0.6 |
| Lung infection | 2 | 0.5 | 2 | 1.1 | 0.3 | 0.0–0.6 |
| Metabolic acidosis | 2 | 0.5 | 2 | 1.1 | 0.3 | 0.0–0.6 |
| Myocardial infarction | 2 | 0.5 | 2 | 1.1 | 0.3 | 0.0–0.6 |
| Nausea | 2 | 0.5 | 2 | 1.1 | 0.3 | 0.0–0.6 |
| Sinus bradycardia | 2 | 0.5 | 2 | 1.1 | 0.3 | 0.0–0.6 |
| Thrush | 2 | 0.5 | 2 | 1.1 | 0.3 | 0.0–0.6 |
| Ventricular arrhythmia | 2 | 0.5 | 2 | 1.1 | 0.3 | 0.0–0.6 |
| Wound infection | 2 | 0.5 | 2 | 1.1 | 0.3 | 0.0–0.6 |
| Anal fistula | 1 | 0.3 | 1 | 0.6 | 0.1 | 0.0–0.4 |
| Arachnoiditis | 1 | 0.3 | 1 | 0.6 | 0.1 | 0.0–0.4 |
| Blood bilirubin increased | 1 | 0.3 | 1 | 0.6 | 0.1 | 0.0–0.4 |
| Bronchopulmonary hemorrhage | 1 | 0.3 | 1 | 0.6 | 0.1 | 0.0–0.4 |
| Calcinosis cutis | 1 | 0.3 | 1 | 0.6 | 0.1 | 0.0–0.4 |
| Cardiac dysautonomia | 1 | 0.3 | 1 | 0.6 | 0.1 | 0.0–0.4 |
| Conjunctival hemorrhage | 1 | 0.3 | 1 | 0.6 | 0.1 | 0.0–0.4 |
| Encephalitis infection | 1 | 0.3 | 1 | 0.6 | 0.1 | 0.0–0.4 |
| Fibrinogen decreased | 1 | 0.3 | 1 | 0.6 | 0.1 | 0.0–0.4 |
| Headache | 1 | 0.3 | 1 | 0.6 | 0.1 | 0.0–0.4 |
| Hearing impaired | 1 | 0.3 | 1 | 0.6 | 0.1 | 0.0–0.4 |
| Hemorrhagic shock | 1 | 0.3 | 1 | 0.6 | 0.1 | 0.0–0.4 |
| Ileal perforation | 1 | 0.3 | 1 | 0.6 | 0.1 | 0.0–0.4 |
| Lymphocyte count decreased | 1 | 0.3 | 1 | 0.6 | 0.1 | 0.0–0.4 |
| Respiratory depression | 1 | 0.3 | 1 | 0.6 | 0.1 | 0.0–0.4 |
| Rinovirus infection | 1 | 0.3 | 1 | 0.6 | 0.1 | 0.0–0.4 |
| Sinus tachycardia | 1 | 0.3 | 1 | 0.6 | 0.1 | 0.0–0.4 |
| Varicella | 1 | 0.3 | 1 | 0.6 | 0.1 | 0.0–0.4 |

Abbreviations. CI: confidence interval.

**Table 5. Severity of adverse events and their causes.**

| Type of adverse event | Total (n = 399) | Severity–n (%) | | | | |
|---|---|---|---|---|---|---|
| | | Mild (n = 57) | Moderate (n = 132) | Severe (n = 128) | Life-threatening (n = 64) | Death-related (n = 18) |
| **Drugs** | 367 | 56 (98.2) | 126 (95.5) | 112 (87.5) | 57 (89.1) | 16 (88.9) |
| Adverse drug reaction | 340 | 56 (98.2) | 116 (87.9) | 103 (80.5) | 51 (79.7) | 14 (77.8) |
| Medication error | 27 | 0 | 10 (7.6) | 9 (7.0) | 6 (9.4) | 2 (11.1) |
| **Hospital care** | 19 | 1 (1.8) | 3 (2.3) | 12 (9.4) | 2 (3.1) | 1 (5.6) |
| Hospital-acquired infections | 17 | 1 (1.8) | 2 (1.5) | 11 (8.6) | 2 (3.1) | 1 (5.6) |
| Others | 2 | 0 | 1 (0.8) | 1 (0.8) | 0 | 0 |
| **Procedures** | 13 | 0 | 3 (2.3) | 4 (31) | 5 (7.8) | 1 (5.6) |
| Lumbar puncture | 11 | 0 | 3 (2.3) | 3 (2.3) | 5 (7.8) | 0 |
| Catheter insertion | 2 | 0 | 0 | 1 (0.8) | 0 | 1 (5.6) |

AEs, with 122 (58.1%) cases, including 75 cases of febrile neutropenia (35.7%), 24 cases of sepsis (11.4%) and 14 HAIs (6.7%) (S4 Table).

As for care necessities, 181 AEs (45.4%) required minimal interventions or outpatient follow-up for their management, 150 AEs (37.6%) resulted in 138 hospital readmissions involving 94 patients (51.9%), and 68 AEs (17.0%) prolonged hospital stay. Four patients required surgical interventions. Each readmission for AEs had a mean stay of 12.6 days (median: 10.5 [6–14] days). Of these readmissions, 58.6% were caused by infections as febrile neutropenia (42.7%) and sepsis (8.7%), while the main non-infectious causes of readmission were mucositis (8.7%), stroke (5.3%), vomiting (4.7%) and paralytic ileus (3.3%) (S5 Table).

Regarding fatal AEs, infectious processes constituted the cause of death in eight cases (44.4%), followed by pancreatitis in three (16.7%) and stroke in two (11.1%) (S6 Table).

Finally, not all AEs occurred independently. On 22 occasions, two or more AEs occurred secondarily to a primary AE, involving 51 AEs (12.8%). Eight of these processes with multiple sequential AEs ended up in patient death. In thirteen cases with multiple AEs, grade 4 was the highest severity, but the patients were able to recover.

## Preventability

During induction, 42 AEs (10.5%) were preventable; 177 (44.4%), ameliorable; and 157 (39.3%), non-preventable. Table 2 shows the preventability rates. In particular, the preventable fraction of AEs that required hospital readmission was higher than the preventable fraction in AEs with outpatient management, being 15.6% versus 4.4%, respectively. This accounted for 9.4 [95% CI: 6.2–12.5] preventable AEs per 1000 hospital patient-days. In addition, severe AEs accounted for 54.8% of preventable and 42.4% of ameliorable AE, respectively. On the other hand, moderate and mild AEs together represented 63.0% of non-preventable AEs in the cohort (Table 6). Specific preventability rates by degree of severity in Fig 1.

**Table 6. Adverse events preventability and severity during induction.**

| Severity | Total (n = 399) | Preventability–n (%) | | | |
|---|---|---|---|---|---|
| | | Preventable (n = 42) | Ameliorable (n = 177) | Non-preventable (n = 157) | Non-evaluable (n = 23) |
| Mild | 57 | 0 | 6 (3.4) | 36 (22.9) | 15 (65.2) |
| Moderate | 132 | 8 (19.0) | 56 (31.6) | 63 (40.1) | 5 (21.7) |
| Severe | 128 | 23 (54.8) | 75 (42.4) | 27 (17.2) | 3 (13.0) |
| Life-threatening | 64 | 8 (19.0) | 33 (18.6) | 23 (14.6) | 0 |
| Death-related | 18 | 3 (7.1) | 7 (4.0) | 8 (5.1) | 0 |

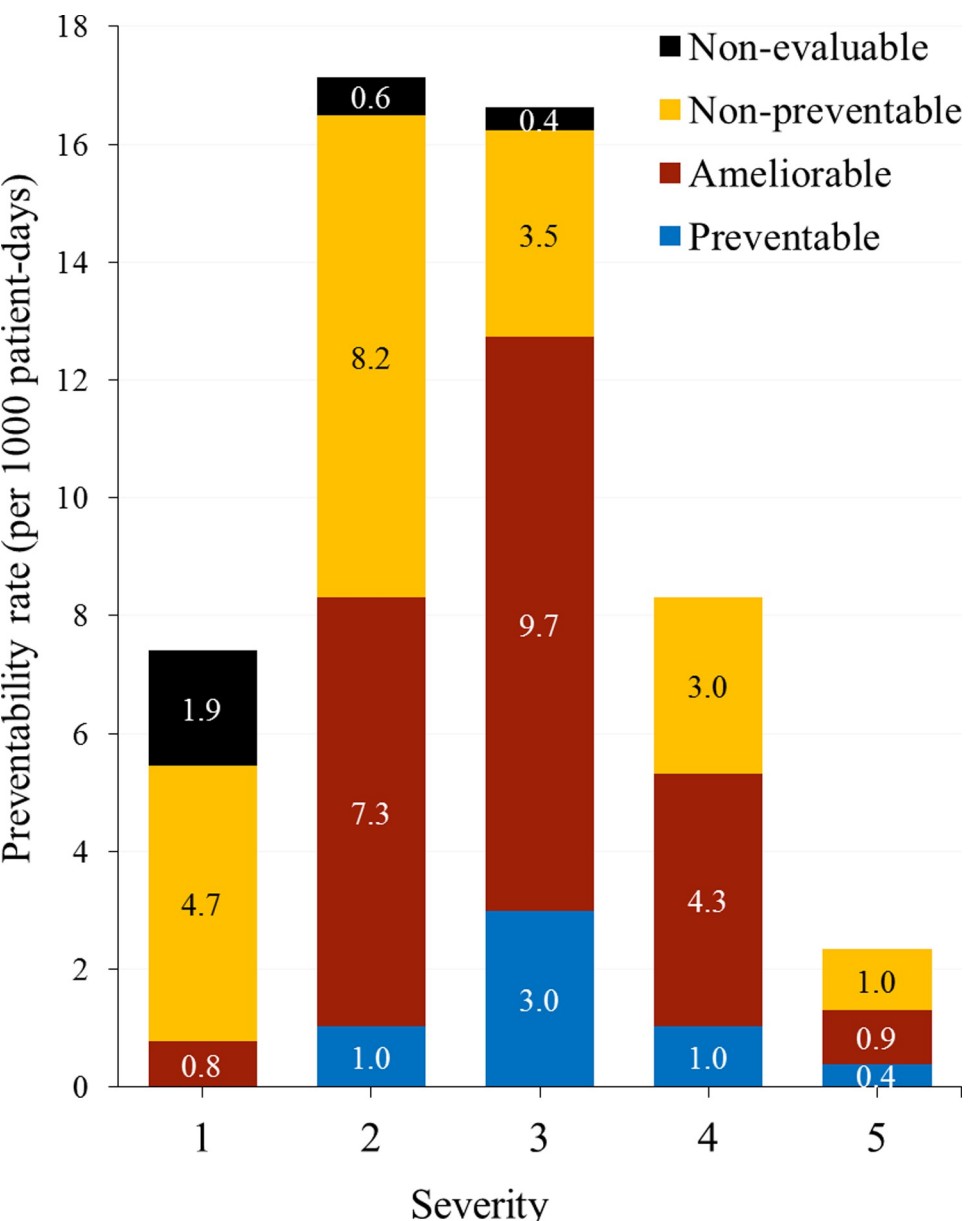

**Fig 1. Preventability of adverse events by severity grade during induction.**

When preventable and ameliorable AEs were considered together, infections accounted for 121 AEs (55%), reaching a rate of 15.7 [95% CI: 12.9–18.5] preventable or ameliorable infections per 1000 patient-days. In contrast, allergic reactions, hyperglycemia, and peripheral neuropathy were the most common non-preventable AEs, accumulating together 69 AEs (43.9%), which is equivalent to 9.0 [95% CI: 6.8–11.1] non-preventable AEs per 1000 patient-days.

Three deaths were considered preventable by the reviewers: one associated with chemotherapy overdose, one with HAI, and the third one was related to central catheter insertion, accounting for 15.0% of deaths during induction (Table 6 and S7 Table). Other seven fatal cases were considered ameliorable: in three, there was evidence of an overdose of any of the chemotherapeutic drugs, without this being enough to have caused the deaths and, in four

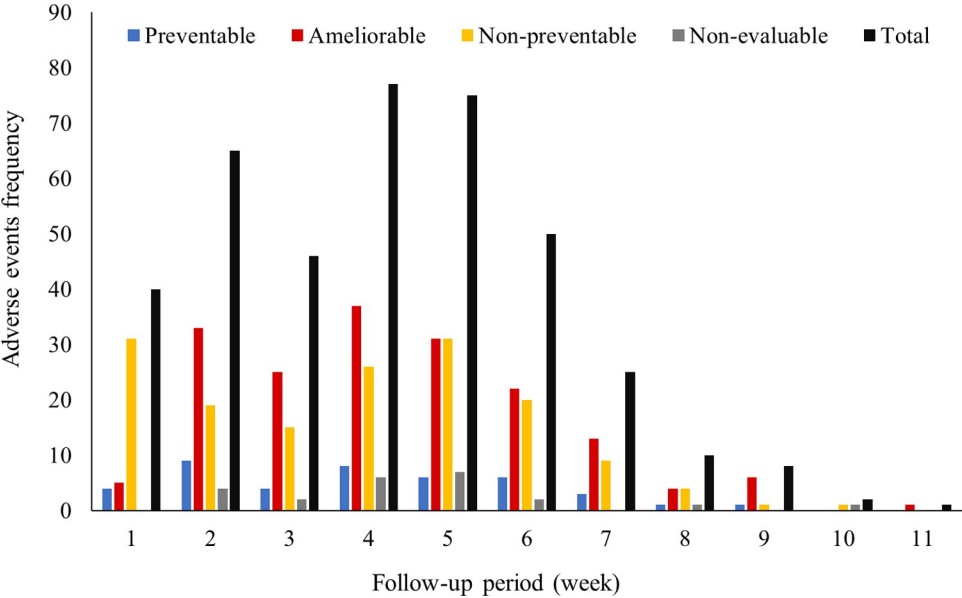

**Fig 2. Frequency of adverse events per follow-up week.**

cases, the originating AEs were community-acquired infections, the prevention of which does not only involve medical actions.

Throughout the follow-up period, the fourth and fifth weeks had the highest number of AEs. From the start of the steroid window (week 2) and induction chemotherapy (weeks 3–6), more than three quarters of total AEs (78.4%) were accumulated (Fig 2). Preventable AEs had a higher occurrence on weeks two and four, particularly ME-related ADEs. At sixth week of follow-up, preventable AEs were HAIs and evaluation medical procedures. In turn, allergic reactions to drugs or hyperglycemia secondary to steroids predominated on first two weeks of follow-up, while febrile neutropenia and myelosuppression were more common between the second and fourth weeks. Some other ADEs predominated between the third and fourth week of induction (fifth and sixth of follow-up) such as strokes and seizures mainly associated with L-asparaginase, or paralytic ileus due to vincristine (S8 Table).

## Discussion

This is the first study in a pediatric National Health Institute in Mexico to comprehensively describe the incidence, severity and preventability of AEs occurring during the care of a group of patients diagnosed with *de novo* ALL until the end of remission induction. It constitutes part of a project focused on the safety and quality of the pediatric oncology care processes at the HIMFG.

In the literature, estimated rates of AEs in cancer patients from different countries range from 2.3 to 96.5 AEs per 1000 patient-days [10, 27, 34]. Although the overall incidence rate observed in this study falls right in the middle of that range, no direct comparison between them is appropriate.

The fraction of AEs considered preventable in this study was lower than reported in other studies with cancer patients with different preventability scales, whose proportion ranges from 22% to 36.3% [4, 10, 27]. In general, the incidence of preventable AEs in pediatric hospitals is low [35], but the incidence rate of ADR in a short period, specially related to chemotherapy, does not allow us to observe the real magnitude of preventable AEs. For example, the rate of

preventable ADEs in pediatric hospitals ranges from 0.5 a 3.8 per 100 admissions [35], while the present study observed 9.1 preventable ADE per 100 admissions. However, there are other difficulties in assessing the preventability of an AE, for example: children with LLA in induction therapy to remission, not only experience multiple effects of toxicity to chemotherapy while the decease is clinically active, but undergo invasive procedures for its diagnosis and subsequent management, therefor, except for catastrophic errors, it is not always easy to evaluate the outcome of a counterfactual hypothesis for specific cases.

It should be noted that most of the AEs considered preventable where related to MEs, HAI or lumbar punctures, sharing their in-hospital nature. In this cohort ME-related ADEs were shown to account for 6.8% of total AEs during induction. A previous study with Mexican children with ALL showed that 47.5% of patients in the induction phase had one or more dosing errors in chemotherapy, producing ADEs in 12.2% of patients. There, a security gap related to medical prescription and the lack of effective barriers to intercept potentially harmful MEs through the medication system were identified [18]. It should also be considered that this work only identified the harm of MEs originated in the hospital but not those originated at home, therefor, the incidence of ADEs related to MEs could actually be underestimated: a study in the United States observed a rate of 3.4 ME-related ADEs per 100 outpatient visits of children with ALL, where most of these originated at home [33].

There is limited evidence on organizational interventions focused on MEs harm reduction in pediatric hospitals: the absolute reduction of MEs may have a modest effect by decreasing hospital stay without reducing costs, and furthermore harm from ADEs is not necessarily reduced [36]. Likewise, the involvement of children´s caregivers in the identification of harmful or potentially harmful MEs can increase the probability of intercepting them by 1.3 times [37].

Other relevant aspect was that 53% of the children in this cohort were readmitted to the hospital due to some AE after induction was started, while one study in the United States has reported a rate of readmission of 36% by AEs in pediatric patients with ALL on induction [38]. It should be noted the higher frequency of hospitalization and longer average stay (12.6 days), may translate into higher complexity in the management of AEs, greater use of resources for care, and increased risk of HAI, as well as a heavy physical and emotional burden for the children [3, 5] and their parents at this first stage [39, 40]. In addition, these AEs represented causes for chemotherapy interruption, which occurs in 11.2% of patients during induction in Mexico [41].

In the present study, febrile neutropenia and other infectious processes were the main causes of hospital readmission, as well as the largest source of ameliorable and preventable AEs. AEs associated with infections predominate during pediatric ALL treatment early phases [42–44]. A pharmacovigilance study showed that febrile neutropenia and sepsis were the most reported ADEs among Mexican pediatric patients with neoplasms [9]. It should be noted that socioeconomic inequalities in low- and middle-income countries favor exposure to infections in the most impoverished patients [6, 8]. But access to more efficacious prophylactic measures against infections and adherence at home is also required [42]. Unfortunately, other types of therapeutic interventions available to ameliorate ADEs are only accessible at an extremely high cost for health systems with limited resources, and even if the frequency and number of readmissions associated with infections were reduced, these interventions have not been shown to improve overall mortality [5].

Another topic for discussion is the occurrence of three theoretically preventable AE-associated deaths: one related to HAI, another to a dosing error and the other to a medical procedure. In this regard, it has been pointed out that cancer patients are at higher risk of HAI, surgical complications or ADEs than other non-cancer patients [9, 34]. Furthermore, in this

cohort, HAIs constituted the main AE associated with hospital care. If we consider that infection-related mortality in children with ALL in high-income countries is close to 1% [43, 44], while in this and other studies in Mexico it ranges from 1.3% to 6.4% [41, 45], infection-related mortality clearly continues to be a barrier to the reduction of morbidity and mortality in children with ALL.

The usefulness of this type of evaluations on the safety and quality of the care processes lies not only in indicating the nature and extent of harm to the patient from the perspective of multiple aspects of care, but it also helps to identify aspects where there are opportunities for improvement, to generate indicators and to take actions in favor of patients who receive medical care [2]. In this sense, the HIMFG has recently adopted initiatives to improve the quality of care and patient safety, among which is an electronic medical record with a computerized physician order entry system, However, this and future interventions must also be evaluated to ensure that better patient safety has been achieved.

Observations in this study have various limitations. The first one involves the quality and completeness of the medical records available for this study, since the records of 11 eligible patients were not available for review and inclusion in the cohort. In addition, retrospective studies may underestimate the frequency of AEs, and AEs incidence rates should therefore be compared with future prospective studies.

In second place, pediatric oncologists had an acceptable agreement in the assessment of AEs preventability, but the largest disagreements in the preventability evaluation occurred between AEs that were considered non-preventable by one reviewer and ameliorable by another in the first round. This discrepancy occurred because the definition of "ameliorable" AE may refer to unavoidable AEs whose severity might have been reduced or improved by better practices or additional interventions. As in other studies, the definition may affect the reproducibility of the preventability of AEs [30, 46]. In this case, such ambiguity in definition may skew the assessment based on severity; it is possible to observe an association between preventability/mitigation and severity degree $\geq 3$, especially when transforming the variables to dichotomous scale (not shown). However, this may also reflect true patterns in attention to some AEs, for example: the infrequent antiemetic premedication for chemotherapy during induction when considering that L-asparaginase y vincristine have low emetogenic potential, so vomiting was classified as mitigable in most cases.

Another important limitation is that the results are only valid for the induction phase in children with ALL, which does not represent the totality of the care processes within the HIMFG and, in addition, current context of care might differ. Therefore, these results should be interpreted considering the context. Finally, this work did not address patient factors, or the type of leukemia associated with the presence of AEs because these issues will be the subject of a subsequent analysis.

## Conclusions

Pediatric patients with ALL experience a large amount of AEs from diagnosis to induction completion with an estimated rate of 51.8 AEs per 1000 patient-days. ADEs accounted for 92% of EAs and were the main ones in frequency, severity and opportunities for prevention or amelioration. In addition, community-acquired infections were the most relevant ameliorable or preventable AEs, given that they were the main cause of hospital readmissions. Finally, the results suggest that an improvement in the safety of medication processes and prevention of hospital-acquired infections might contribute to the reduction of early mortality in pediatric ALL.

## Supporting information

**S1 Table. Incidence of adverse events by patients and admissions during induction therapy.**
(DOCX)

**S2 Table. Individual description of adverse drug events occurred during remission induction.**
(DOCX)

**S3 Table. Adverse events incidence rate by system organ class during induction.**
(DOCX)

**S4 Table. Adverse events with severity grade ≥3 observed during induction.**
(DOCX)

**S5 Table. Adverse events that required hospital admission during induction.**
(DOCX)

**S6 Table. Adverse events frequency by severity and preventability during induction.**
(DOCX)

**S7 Table. Preventability of adverse events and their causes.**
(DOCX)

**S8 Table. Adverse events weekly frequency during remission induction.**
(DOCX)

**S1 File. Research protocol.**
(PDF)

**S2 File. Letter of approval from the institutional review board.**
(PDF)

**S1 Dataset. Patient data.**
(XLSX)

**S2 Dataset. Adverse event data.**
(XLSX)

## Author Contributions

**Conceptualization:** Edmundo Vázquez-Cornejo, Olga Morales-Ríos.

**Data curation:** Edmundo Vázquez-Cornejo.

**Formal analysis:** Edmundo Vázquez-Cornejo, Gabriela Hernández-Pliego, Carlo Cicero-Oneto.

**Investigation:** Edmundo Vázquez-Cornejo, Gabriela Hernández-Pliego.

**Methodology:** Edmundo Vázquez-Cornejo, Olga Morales-Ríos, Juan Garduño-Espinosa.

**Project administration:** Olga Morales-Ríos, Juan Garduño-Espinosa.

**Supervision:** Juan Garduño-Espinosa.

**Validation:** Olga Morales-Ríos, Gabriela Hernández-Pliego, Carlo Cicero-Oneto.

**Writing – original draft:** Edmundo Vázquez-Cornejo.

**Writing – review & editing:** Edmundo Vázquez-Cornejo, Olga Morales-Ríos, Gabriela Hernández-Pliego, Carlo Cicero-Oneto, Juan Garduño-Espinosa.

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
