## [Decision Letter · Decision Letter 0]

4 Jan 2022

PONE-D-21-37347Incidence, severity, and preventability of adverse events during the induction of patients with Acute Lymphoblastic Leukemia in a tertiary care pediatric hospital in Mexico.PLOS ONE

Dear Dr. Vázquez-Cornejo,

Thank you for submitting your manuscript to PLOS ONE. After careful consideration, we feel that it has merit but does not fully meet PLOS ONE’s publication criteria as it currently stands. Therefore, we invite you to submit a revised version of the manuscript that addresses the points raised during the review process.

We look forward to receiving your revised manuscript.

Kind regards,

Paula Schaiquevich

Academic Editor

PLOS ONE

Journal Requirements:

3.  We noted in your submission details that a portion of your manuscript may have been presented or published elsewhere. [This manuscript shares data from a previous publication focused on the identification of medication errors to chemotherapy in the medication system of our Hospital (a copy is attached). Variables of baseline characteristics of the sample are shared, specifically: age, sex, leukocytes, risk classification, immunophenotype, comorbidities and nutritional status, all of which are contained in Table 1.

Also, the medication errors shown in Table 3 of the current manuscript contain the chemotherapy errors already published, but are not limited to them, since the current data covers other non-chemotherapeutic pharmacological therapies.

We consider that the data mentioned here do not constitute a duplicity, because both studies differ in purpose: the first describes in detail elements of the process (medication errors) of a particular type of medical act (medication with chemotherapy). While the other is a comprehensive description of the outcome variables (adverse events) of all the care processes (drugs, general hospital care, and medical or surgical procedures) on a clinical group of great interest to our Institution, with a focus on the preventability. In accordance with the above, baseline cohort data are not the primary endpoints in either study, but they allow us to contextualize the target population. Furthermore, the results reported in this manuscript cannot be deduced from the previous publication or vice versa.

In summary, the related publication and this manuscript answer different research questions, and the data that are related are not the primary variable in any of the studies.] 

Please clarify whether this publication was peer-reviewed and formally published. If this work was previously peer-reviewed and published, in the cover letter please provide the reason that this work does not constitute dual publication and should be included in the current manuscript.

Additional Editor Comments:

After careful examination, the manuscript entitled "Incidence, severity, and preventability of adverse events during the induction of patients with Acute Lymphoblastic Leukemia in a tertiary care pediatric hospital in Mexico." could be considered for publication. Please, check for the comments of the reviewers and clearly describe the adverse events that were recorded in relation to the agents administered.

Reviewers' comments:

Reviewer's Responses to Questions

**Comments to the Author**

1. Is the manuscript technically sound, and do the data support the conclusions?

Reviewer #1: Yes

Reviewer #2: Yes

2. Has the statistical analysis been performed appropriately and rigorously? 

Reviewer #1: Yes

Reviewer #2: Yes

3. Have the authors made all data underlying the findings in their manuscript fully available?

Reviewer #1: Yes

Reviewer #2: Yes

4. Is the manuscript presented in an intelligible fashion and written in standard English?

Reviewer #1: Yes

Reviewer #2: Yes

5. Review Comments to the Author

Reviewer #1: The research presents a good clinical-administrative method to generate indicators of adverse reactions of oncological drugs for the treatment of ALL in pediatrics, the way it presents the results is innovative and can generate a new method for the monitoring of adverse reactions, it is important to be able to Generate a table in the manuscript about the drug individually to generate the adverse reaction and also in which part of the protocol this risk is increased the most and from that action to be able to generate a prevention method.

Perhaps adding a previously defined data will provide more robustness to the research generated.

Reviewer #2: The work should contain a table describing all the adverse effects recorded. It would have to include the severity of the same and the causality reached based on some international algorithm, where it is clearly identifying which drug of the induction each of them correspond to.

6. PLOS authors have the option to publish the peer review history of their article (what does this mean?). If published, this will include your full peer review and any attached files.

Reviewer #1: **Yes: **Jorge Morales Vallespín

Reviewer #2: No

---

## [Author Response · Author response to Decision Letter 0]

26 Feb 2022

Mexico City, Mexico, February 25, 2022

Dr. Emily Chenette

Editor-in-Chief

PLOS ONE

Dear editor:

On behalf of all the authors, I thank you for having considered the manuscript PONE-D-21-37347 “Incidence, severity, and preventability of adverse events during the induction of patients with acute lymphoblastic leukemia in a tertiary care pediatric hospital in Mexico”, which was submitted for review at PLOS ONE as a Research Article.

Below you will find our response to the points mentioned by the Academic Editor and the Reviewers in the decision letter received on January 4, 2022.

(A) Journal Requirements

Comment 1. Please ensure that your manuscript meets PLOS ONE's style requirements, including those for file naming. The PLOS ONE style templates can be found at 

Response. We appreciate this observation because it has allowed us to make style corrections throughout the revised manuscript. After reviewing the style requirements for the revised files, the following elements were modified: 

Title page: format and content were reviewed in the revised manuscript. There, the use of capital letters, punctuation, and affiliations that designate positions in the institution were corrected. But we couldn't find the font size for the title, and thus we chose the size that was closest to the one on the sample page.

Abstract: an abbreviation was replaced by the corresponding text. In addition, internal segments were eliminated and the text was unified in a single paragraph, following the submission guidelines and the sample page offered by the Academic Editor. 

Materials and methods: the heading of the “Methods” section of the original manuscript was modified in the revised manuscript, now it reads “Materials and methods”, according to the guidelines and format sample offered by PLOS ONE.

Tables. The framework of the tables included in the manuscript was revised to meet the required parameters. Only the sample sizes "n" were put in brackets to avoid double-rowing in cells with headings.

References: after reviewing the numerical sequence of the references in the text, references #42, #45 and #36 (lines 350-354, original manuscript) were corrected, replacing them with the correct numerals in the revised manuscript, which now are #36 , #37 and #38, respectively (lines 378-382, revised manuscript). The contents in said paragraphs or references was not modified. 

In addition, we verified compliance with the style rules for references, including the validity of links associated with electronic sources. Therefore, the citation date was updated for references #13, #19, and #20. In addition, format adaptations were made to the pagination numbers of the following references: #2, #3, #5, #15 (additionally, original language was mentioned), #16, #18, #27, #38 (in addition, volume with supplement format was corrected), #42, #43. 

Only two references required an update: in reference #13, the original electronic link now leads to an updated version of 04 January 2022, which modifies the title of the 2021 version in PubMed, as well as the authors; original citation was: Rodziewicz TL, Hipskind JE. Medical Error Prevention. [Updated 2021 Jan 4]. In: StatPearls [Internet]. Treasure Island (FL): StatPearls Publishing; 2021 [cited 2021 Feb 15]. Available from: https://www.ncbi.nlm.nih.gov/books/NBK499956/. 

Now, the updated citation (including the style) in the revised manuscript is: 

Rodziewicz TL, Houseman B, Hipskind JE. Medical Error Reduction and Prevention. 2022 Jan 4 [cited 2022 Feb 02]. In: StatPearls [Internet]. Treasure Island (FL): StatPearls Publishing; c2022. Available from: https://www.ncbi.nlm.nih.gov/books/NBK499956/

In addition, the main authors were added in reference #1. In the original manuscript it appears as: World Health Organization‎. World alliance for patient safety: WHO draft guidelines for adverse event reporting and learning systems: from information to action. Geneva: World Health Organization; 2005. Document: WHO/EIP/SPO/QPS/05.3. [cited 2022 Feb 02] Available from: https://apps.who.int/iris/handle/10665/69797

The citation, with final format, now is: 

Leape L, Abookire S. World alliance for patient safety: WHO draft guidelines for adverse event reporting and learning systems: from information to action. Geneva (GVA): World Health Organization; 2005. Document No.: WHO/EIP/SPO/QPS/05.3. Available from: https://apps.who.int/iris/handle/10665/69797

Supporting information: in the supporting information, the labels of all the files attached to the revised manuscript were reviewed, with the names of the files previously identified as “Appendix” being changed into “File”, since it was considered more appropriate for those types of files. In addition, the databases contained in the “S3 Appendix. Databases” file were separated and sent individually as “S1 Dataset” and “S2 Dataset”, for greater clarity in content identification.

The guidelines for “Financial Disclosure”, “Competing Interests” and “Data Availability” statements were reviewed, and maintaining them unchanged was considered appropriate, given that they meet PLOS ONE explicit requirements.

Comment 2. Please provide additional details regarding participant consent. In the ethics statement in the Methods and online submission information, please ensure that you have specified what type you obtained (for instance, written or verbal, and if verbal, how it was documented and witnessed). If your study included minors, state whether you obtained consent from parents or guardians. If the need for consent was waived by the ethics committee, please include this information.

Response. With regards to this requirement, the ethical approval section in the revised manuscript has been expanded. A broader description of the research work approval process and the reasons why the Ethics Committee at our institution allowed a waiver of written informed consent for the conduction of this study were provided. The extension reads as follows (page 4-5, lines 80-93, revised manuscript):

The HIM-2021-065 study protocol was submitted for review to the Research, Research Ethics, and Biosafety Committees of the Federico Gómez Children’s Hospital of Mexico (HIMFG – Hospital Infantil de México Federico Gómez). The Research Ethics Committee evaluated the study design and the sources of information for this work, which involved a retrospective review of medical records generated during routine patient care and databases used for pharmacovigilance analysis, which is why this study did not involve the performance of medical interventions or intentional modification of physiological or psychological variables by the researchers on patients or their legal guardians. Therefore, the Institutional Research Ethics Committee classified this project as research free of risk for the study subjects, with approval of the protocol being granted. In addition, the Research Ethics Committee exempted the researchers from obtaining informed consent, emphasizing the obligation of the researchers regarding the protection of the identity and privacy of the patients included in the study in order for sensitive information that might lead to patient identification not to be disclosed in this publication, in the supporting material or in the databases that are made available to the public for consultation.

Comment 3. We noted in your submission details that a portion of your manuscript may have been presented or published elsewhere. [This manuscript shares data from a previous publication focused on the identification of medication errors to chemotherapy in the medication system of our hospital (a copy is attached). Variables of baseline characteristics of the sample are shared, specifically: age, sex, leukocytes, risk classification, immunophenotype, comorbidities and nutritional status, all of which are contained in Table 1.

Also, the medication errors shown in Table 3 of the current manuscript contain the chemotherapy errors already published, but are not limited to them, since the current data covers other non-chemotherapeutic pharmacological therapies.

We consider that the data mentioned here do not constitute a duplicity, because both studies differ in purpose: the first describes in detail elements of the process (medication errors) of a particular type of medical act (medication with chemotherapy). While the other is a comprehensive description of the outcome variables (adverse events) of all the care processes (drugs, general hospital care, and medical or surgical procedures) on a clinical group of great interest to our Institution, with a focus on the preventability. In accordance with the above, baseline cohort data are not the primary endpoints in either study, but they allow us to contextualize the target population. Furthermore, the results reported in this manuscript cannot be deduced from the previous publication or vice versa.

In summary, the related publication and this manuscript answer different research questions, and the data that are related are not the primary variable in any of the studies.]

Please clarify whether this publication was peer-reviewed and formally published. If this work was previously peer-reviewed and published, in the cover letter please provide the reason that this work does not constitute dual publication and should be included in the current manuscript.

Response. Understanding the importance of this observation, the relevant explanation was included both in the cover letter and in the revised manuscript. In both additions, the fact that the publication in question is formally published and was reviewed by peers is clarified, as well as the reasons why that study and the present one do not constitute duplicate publications.

An additional paragraph was included in the revised manuscript, in the ethical approval section, explaining the role of the data mentioned in the preceding publication and in the study under review. The added paragraph is the following (Page 5, lines 94-102, revised manuscript): 

In a previous peer-reviewed publication, we used information on chemotherapy from this cohort to study MEs. We emphasize that the study focused on MEs and the present work on AEs have different purposes: the former focused on the quality of the chemotherapy medication process and the current work focuses on the preventability of AEs related to all healthcare-related interventions (drugs, procedures and hospital care). Since the primary variables in both studies are conceptually and operationally different, there is no overlap or duplication in the results. Finally, demographic data such as gender, age, risk classification, comorbidities and nutritional status shown in the previous publication have been included to contextualize the target population, without them being primary variables in the study.

The corresponding explanation is also included in the cover letter, with the digital object identifier (Doi) of the formal publication to which reference is made being included. The updated text is the following:

In a previous peer-reviewed publication (doi: 10.1002/cam4.2438, a copy of which is attached to the Editorial Manager as “Related Manuscript file type”), we used information on chemotherapy from this cohort to study medication errors. However, we emphasize that the study on medication errors and current manuscript on adverse events have different purposes and scopes: the former focused on the quality of the medication process in chemotherapy, and current work focuses on the preventability of harm related to all healthcare-related interventions (drugs, procedures and hospital care). Since the primary variables in both studies are conceptually and operationally different, there is no overlap or duplication in the results. Finally, demographic data such as gender, age, risk classification, comorbidities and nutritional status shown in the previous publication have been included to contextualize the target population, without them being primary variables in the study.

Comment 4. Please review your reference list to ensure that it is complete and correct. If you have cited papers that have been retracted, please include the rationale for doing so in the manuscript text, or remove these references and replace them with relevant current references. Any changes to the reference list should be mentioned in the rebuttal letter that accompanies your revised manuscript. If you need to cite a retracted article, indicate the article’s retracted status in the References list and also include a citation and full reference for the retraction notice.

Response. The list of references that were included in the first version of the manuscript was exhaustively reviewed; for this, information from PubMed, the journals that host each cited article, and web tools such as Retraction Watch Database and Scite were used. In the search, no retracted articles or papers with errata were identified. Despite not having found retracted citations in our review, we are at PLOS ONE Academic Editor disposal to correct any element in the list of the revised manuscript that warrants it and that might have been overlooked.

(B) Reviewers' comments.

Comments to the Author:

Reviewer #1: The research presents a good clinical-administrative method to generate indicators of adverse reactions of oncological drugs for the treatment of ALL in pediatrics, the way it presents the results is innovative and can generate a new method for the monitoring of adverse reactions, it is important to be able to Generate a table in the manuscript about the drug individually to generate the adverse reaction and also in which part of the protocol this risk is increased the most and from that action to be able to generate a prevention method.

Perhaps adding a previously defined data will provide more robustness to the research generated.

Response. We thank Reviewer #1 for his observations on this work. In response, a table with a complete list of adverse drug reactions identified in the study was added, noting adverse drug reactions and adverse drug events related to medication errors according to the definitions in our manuscript. This new table included the individual term (these terms are compatible with MedDRA’s LTTs), the drug (or combination of drugs) suspected to be the causal agent, causality assigned by the Naranjo Algorithm (required by the official Mexican standard for pharmacovigilance) and the severity of each event.

Due to the length of the table, including it as supplementary material was decided, under the name “S2 Table. Individual description of adverse drug events occurred during remission induction”. In the revised manuscript, this new table is mentioned in the “Results” section, lines 245-246, in the following text: 

In general, ADEs were the most frequent, accounting for 92.0% of total AEs (complete list in S2 Table).

Furthermore, to illustrate the frequency of adverse events throughout the study period, showing a figure was decided (Figure 2. Adverse events weekly distribution), including all identified adverse events separated by preventability category. The most common ADEs per week of the induction protocol are also described (S8 Table. Adverse events weekly frequency during remission induction). Both these new elements are mentioned in the “Results” section of the revised manuscript as follows (lines 330-340):

Throughout the follow-up period, the fourth and fifth weeks had the highest number of AEs. From the start of the steroid window (week 2) and induction chemotherapy (weeks 3-6), more than three quarters of total AEs (78.4%) were accumulated (Fig. 2). Preventable AEs had a higher occurrence on weeks two and four, particularly ME-related ADEs. At sixth week of follow-up, preventable AEs were HAIs and evaluation medical procedures. In turn, allergic reactions to drugs or hyperglycemia secondary to steroids predominated on first two weeks of follow-up, while febrile neutropenia and myelosuppression were more common between the second and fourth weeks. Some other ADEs predominated between the third and fourth week of induction (fifth and sixth of follow-up) such as strokes and seizures mainly associated with L-asparaginase, or paralytic ileus due to vincristine (S8 Table).

Reviewer #2: The work should contain a table describing all the adverse effects recorded. It would have to include the severity of the same and the causality reached based on some international algorithm, where it is clearly identifying which drug of the induction each of them correspond to.

Response. We appreciate the observation made by Reviewer #2 asking us to include a more detailed description of the adverse drug reactions identified in the study. For this reason, supplementary material “S2 Table. Individual description of adverse drug events occurred during remission induction” was enriched by including the variables suggested in your kind comment: the causality of each event according to the Narnajo algorithm, which is widely accepted in pharmacovigilance activities. In addition, the drug or combination of drugs corresponding to each adverse drug reaction was identified. Finally, an assessment of the severity reached by each event was also included in the same, above-mentioned supplementary file.

Finally, we hope that the changes made to the revised manuscript respond to the observations and requirements presented by the Editor and both Reviewers, and achieve an adequate form that is suitable for publication in PLOS ONE. 

We look forward to hearing from you in due time regarding our submission and to respond to any further questions and comments you may have.

On behalf of the authors of this work, we extend you a warm greeting.

Edmundo Vázquez Cornejo, M.Sc.

Corresponding author

edmundoepiclin.qfb@outlook.com

Telephone: + (52) 5228 9917 ext. 2356

---

## [Editor Report · Decision Letter 1]

2 Mar 2022

Incidence, severity, and preventability of adverse events during the induction of patients with acute lymphoblastic leukemia in a tertiary care pediatric hospital in Mexico.

PONE-D-21-37347R1

Dear Dr. Vázquez-Cornejo

We’re pleased to inform you that your manuscript has been judged scientifically suitable for publication and will be formally accepted for publication once it meets all outstanding technical requirements.

Kind regards,

Paula Schaiquevich

Academic Editor

PLOS ONE

Additional Editor Comments (optional):

I appreciate the authors responding the comments raised by the reviewers and editor. I have no further comments for the authors.
---

## [Editor Report · Acceptance letter]

16 Mar 2022

PONE-D-21-37347R1 

Incidence, severity, and preventability of adverse events during the induction of patients with acute lymphoblastic leukemia in a tertiary care pediatric hospital in Mexico. 

Dear Dr. Vázquez-Cornejo:

I'm pleased to inform you that your manuscript has been deemed suitable for publication in PLOS ONE. Congratulations! Your manuscript is now with our production department. 

Kind regards, 

on behalf of

Dr. Paula Schaiquevich 

Academic Editor

PLOS ONE